# SELF-ENSEMBLING FOR VISUAL DOMAIN ADAPTATION

**Geoff French, Michal Mackiewicz & Mark Fisher**
School of Computing Sciences
University of East Anglia
Norwich
UK
{g.french,m.mackiewicz,m.fisher}@uea.ac.uk

## ABSTRACT

This paper explores the use of self-ensembling for visual domain adaptation problems. Our technique is derived from the mean teacher variant (Tarvainen & Valpola (2017)) of temporal ensembling (Laine & Aila (2017)), a technique that achieved state of the art results in the area of semi-supervised learning. We introduce a number of modifications to their approach for challenging domain adaptation scenarios and evaluate its effectiveness. Our approach achieves state of the art results in a variety of benchmarks, including our winning entry in the VISDA-2017 visual domain adaptation challenge. In small image benchmarks, our algorithm not only outperforms prior art, but can also achieve accuracy that is close to that of a classifier trained in a supervised fashion.

## 1 INTRODUCTION

The strong performance of deep learning in computer vision tasks comes at the cost of requiring large datasets with corresponding ground truth labels for training. Such datasets are often expensive to produce, owing to the cost of the human labour required to produce the ground truth labels.

Semi-supervised learning is an active area of research that aims to reduce the quantity of ground truth labels required for training. It is aimed at common practical scenarios in which only a small subset of a large dataset has corresponding ground truth labels. Unsupervised domain adaptation is a closely related problem in which one attempts to transfer knowledge gained from a labeled source dataset to a distinct unlabeled target dataset, within the constraint that the objective (*e.g.* digit classification) must remain the same. Domain adaptation offers the potential to train a model using labeled synthetic data – that is often abundantly available – and unlabeled real data. The scale of the problem can be seen in the VisDA-17 domain adaptation challenge images shown in Figure 1. We will present our winning solution in Section 4.2.

Recent work (Tarvainen & Valpola (2017)) has demonstrated the effectiveness of self-ensembling with random image augmentations to achieve state of the art performance in semi-supervised learning benchmarks.

We have developed the approach proposed by Tarvainen & Valpola (2017) to work in a domain adaptation scenario. We will show that this can achieve excellent results in specific small image domain adaptation benchmarks. More challenging scenarios, notably MNIST → SVHN and the VisDA-17 domain adaptation challenge required further modifications. To this end, we developed confidence thresholding and class balancing that allowed us to achieve state of the art results in a variety of benchmarks, with some of our results coming close to those achieved by traditional supervised learning. Our approach is sufficiently flexble to be applicable to a variety of network architectures, both randomly initialized and pre-trained.

Our paper is organised as follows; in Section 2 we will discuss related work that provides context and forms the basis of our technique; our approach is described in Section 3 with our experiments and results in Section 4; and finally we present our conclusions in Section 5.

(a) VisDa-17 training set images; the labeled source domain

(b) VisDa-17 validation set images; the unlabeled target domain

Figure 1: Images from the VisDA-17 domain adaptation challenge

## 2  RELATED WORK

In this section we will cover self-ensembling based semi-supervised methods that form the basis of our approach and domain adaptation techniques to which our work can be compared.

### 2.1  SELF-ENSEMBLING FOR SEMI-SUPERVISED LEARNING

Recent work based on methods related to self-ensembling have achieved excellent results in semi-supervised learning scenarios. A neural network is trained to make consistent predictions for unsupervised samples under different augmentation Sajjadi et al. (2016), dropout and noise conditions or through the use of adversarial training Miyato et al. (2017). We will focus in particular on the self-ensembling based approaches of Laine & Aila (2017) and Tarvainen & Valpola (2017) as they form the basis of our approach.

Laine & Aila (2017) present two models; their Π-model and their temporal model. The Π-model passes each unlabeled sample through a classifier twice, each time with different dropout, noise and image translation parameters. Their unsupervised loss is the mean of the squared difference in class probability predictions resulting from the two presentations of each sample. Their temporal model maintains a per-sample moving average of the historical network predictions and encourages subsequent predictions to be consistent with the average. Their approach achieved state of the art results in the SVHN and CIFAR-10 semi-supervised classification benchmarks.

Tarvainen & Valpola (2017) further improved on the temporal model of Laine & Aila (2017) by using an exponential moving average of the network weights rather than of the class predictions. Their approach uses two networks; a *student* network and a *teacher* network, where the student is trained using gradient descent and the weigths of the teacher are the exponential moving average of those of the student. The unsupervised loss used to train the student is the mean square difference between the predictions of the student and the teacher, under different dropout, noise and image translation parameters.

### 2.2  DOMAIN ADAPTATION

There is a rich body of literature tackling the problem of domain adaptation. We focus on deep learning based methods as these are most relevant to our work.

Auto-encoders are unsupervised neural network models that reconstruct their input samples by first encoding them into a latent space and then decoding and reconstructing them. Ghifary et al. (2016) describe an auto-encoder model that is trained to reconstruct samples from both the source and target domains, while a classifier is trained to predict labels from domain invariant features present in the latent representation using source domain labels. Bousmalis et al. (2016) reckognised that samples from disparate domains have distinct domain specific characteristics that must be represented in the latent representation to support effective reconstruction. They developed a split model that separates the latent representation into shared domain invariant features and private features specific to the source and target domains. Their classifier operates on the domain invariant features only.

Ganin & Lempitsky (2015) propose a bifurcated classifier that splits into label classification and domain classification branches after common feature extraction layers. A gradient reversal layer is

placed between the common feature extraction layers and the domain classification branch; while the domain classification layers attempt to determine which domain a sample came from the gradient reversal operation encourages the feature extraction layers to confuse the domain classifier by extracting domain invariant features. An alternative and simpler implementation described in their appendix minimises the label cross-entropy loss in the feature and label classification layers, minimises the domain cross-entropy in the domain classification layers but *maximises* it in the feature layers. The model of Tzeng et al. (2017) runs along similar lines but uses separate feature extraction sub-networks for source and domain samples and train the model in two distinct stages.

Saito et al. (2017a) use tri-training (Zhou & Li (2005)); feature extraction layers are used to drive three classifier sub-networks. The first two are trained on samples from the source domain, while a weight similarity penalty encourages them to learn different weights. Pseudo-labels generated for target domain samples by these source domain classifiers are used to train the final classifier to operate on the target domain.

Generative Adversarial Networks (GANs; Goodfellow et al. (2014)) are unsupervised models that consist of a generator network that is trained to generate samples that match the distribution of a dataset by fooling a discriminator network that is simultaneously trained to distinguish real samples from generates samples. Some GAN based models – such as that of Sankaranarayanan et al. (2017) – use a GAN to help learn a domain invariant embedding for samples. Many GAN based domain adaptation approaches use a generator that transforms samples from one domain to another.

Bousmalis et al. (2017) propose a GAN that adapts synthetic images to better match the characteristics of real images. Their generator takes a synthetic image and noise vector as input and produces an adapted image. They train a classifier to predict annotations for source and adapted samples alonside the GAN, while encouraing the generator to preserve aspects of the image important for annotation. The model of Shrivastava et al. (2017) consists of a refiner network (in the place of a generator) and discriminator that have a limited receptive field, limiting their model to making local changes while preserving ground truth annotations. The use of refined simulated images with corresponding ground truths resulted in improved performance in gaze and hand pose estimation.

Russo et al. (2017) present a bi-directional GAN composed of two generators that transform samples from the source to the target domain and vice versa. They transform labelled source samples to the target domain using one generator and back to the source domain with the other and encourage the network to learn label class consistency. This work bears similarities to CycleGAN, by Zhu et al. (2017).

A number of domain adaptation models maximise domain confusion by minimising the difference between the distributions of features extracted from source and target domains. Deep CORAL Sun & Saenko (2016) minimises the difference between the feature covariance matrices for a mini-batch of samples from the source and target domains. Tzeng et al. (2014) and Long et al. (2015) minimise the Maximum Mean Discrepancy metric Gretton et al. (2012). Li et al. (2016) described *adaptive batch normalization*, a variant of batch normalization (Ioffe & Szegedy (2015)) that learns separate batch normalization statistics for the source and target domains in a two-pass process, establishing new state-of-the-art results. In the first pass standard supervised learning is used to train a classifier for samples from the source domain. In the second pass, normalization statistics for target domain samples are computed for each batch normalization layer in the network, leaving the network weights as they are.

## 3 METHOD

Our model builds upon the mean teacher semi-supervised learning model of Tarvainen & Valpola (2017), which we will describe. Subsequently we will present our modifications that enable domain adaptation.

The structure of the mean teacher model of Tarvainen & Valpola (2017) – also discussed in section 2.1 – is shown in Figure 2a. The student network is trained using gradient descent, while the weights of the teacher network are an exponential moving average of those of the student. During training each input sample $x_i$ is passed through both the student and teacher networks, generating predicted class probability vectors $z_i$ (student) and $\tilde{z}_i$ (teacher). Different dropout, noise and image translation parameters are used for the student and teacher pathways.

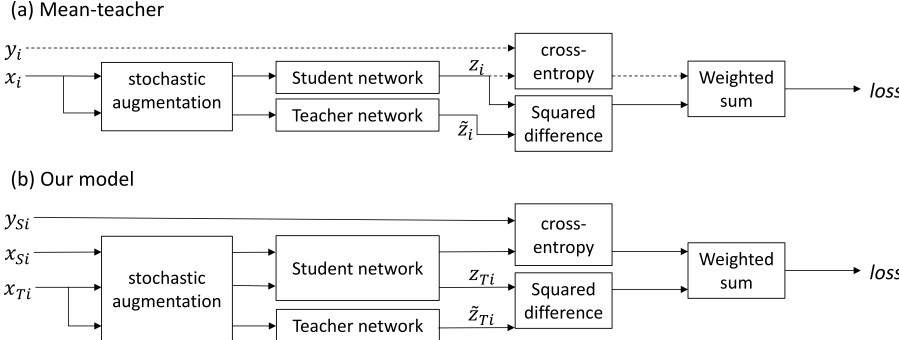

Figure 2: The network structures of the original mean teacher model and our model. Dashed lines in the mean teacher model indicate that ground truth labels – and therefore cross-entropy classification loss – are only available for labeled samples.

During each training iteration a mini-batch of samples is drawn from the dataset, consisting of both labeled and unlabeled samples. The training loss is the sum of a supervised and an unsupervised component. The supervised loss is cross-entropy loss computed using $z_i$ (student prediction). It is masked to 0 for unlabeled samples for which no ground truth is available. The unsupervised component is the self-ensembling loss. It penalises the difference in class predictions between student ($z_i$) and teacher ($\tilde{z}_i$) networks for the same input sample. It is computed using the mean squared difference between the class probability predictions $z_i$ and $\tilde{z}_i$.

Laine & Aila (2017) and Tarvainen & Valpola (2017) found that it was necessary to apply a time-dependent weighting to the unsupervised loss during training in order to prevent the network from getting stuck in a degenerate solution that gives poor classification performance. They used a function that follows a Gaussian curve from 0 to 1 during the first 80 epochs.

In the following subsections we will describe our contributions in detail along with the motivations for introducing them.

## 3.1 Adapting to domain adaptation

We minimise the same loss as in Tarvainen & Valpola (2017); we apply cross-entropy loss to labeled source samples and unsupervised self-ensembling loss to target samples. As in Tarvainen & Valpola (2017), self-ensembling loss is computed as the mean-squared difference between predictions produced by the student ($z_{Ti}$) and teacher ($\tilde{z}_{Ti}$) networks with different augmentation, dropout and noise parameters.

The models of Tarvainen & Valpola (2017) and of Laine & Aila (2017) were designed for semi-supervised learning problems in which a subset of the samples in a single dataset have ground truth labels. During training both models mix labeled and unlabeled samples together in a mini-batch. In contrast, unsupervised domain adaptation problems use two distinct datasets with different underlying distributions; labeled source and unlabeled target. Our variant of the mean teacher model – shown in Figure 2b – has separate source ($X_{Si}$) and target ($X_{Ti}$) paths. Inspired by the work of Li et al. (2016), we process mini-batches from the source and target datasets separately (per iteration) so that batch normalization uses different normalization statistics for each domain during training.[1]. We do not use the approach of Li et al. (2016) as-is, as they handle the source and target datasets separtely in two distinct training phases, where our approach must train using both simultaneously. We also do not maintain separate exponential moving averages of the means and variances for each dataset for use at test time.

---

[1]This is simple to implement using most neural network toolkits; evaluate the network once for source samples and a second time for target samples, compute the supervised and unsupervised losses respectively and combine.

As seen in the 'MT+TF' row of Table 1, the model described thus far achieves state of the art results in 5 out of 8 small image benchmarks. The MNIST → SVHN, STL → CIFAR-10 and Syn-digits → SVHN benchmarks however require additional modifications to achieve good performance.

## 3.2 CONFIDENCE THRESHOLDING

We found that replacing the Gaussian ramp-up factor that scales the unsupervised loss with confidence thresholding stabilized training in more challenging domain adaptation scenarios. For each unlabeled sample $x_{Ti}$ the teacher network produces the predicted class probabilty vector $\tilde{z}_{Tij}$ – where $j$ is the class index drawn from the set of classes $C$ – from which we compute the confidence $\tilde{f}_{Ti} = \max_{j \in C}(\tilde{z}_{Tij})$; the predicted probability of the predicted class of the sample. If $\tilde{f}_{Ti}$ is below the confidence threshold (a parameter search found 0.968 to be an effective value for small image benchmarks), the self-ensembling loss for the sample $x_i$ is masked to 0.

Our working hypothesis is that confidence thresholding acts as a filter, shifting the balance in favour of the student learning correct labels from the teacher. While high network prediction confidence does not guarantee correctness there is a positive correlation. Given the tolerance to incorrect labels reported by Laine & Aila (2017), we believe that the higher signal-to-noise ratio underlies the success of this component of our approach.

The use of confidence thresholding achieves a state of the art results in the STL → CIFAR-10 and Syn-digits → SVHN benchmarks, as seen in the 'MT+CT+TF' row of Table 1. While confidence thresholding can result in very slight reductions in performance (see the MNIST ↔ USPS and SVHN → MNIST results), its ability to stabilise training in challenging scenarios leads us to recommend it as a replacement for the time-dependent Gaussian ramp-up used in Laine & Aila (2017).

## 3.3 DATA AUGMENTATION

We explored the effect of three data augmentation schemes in our small image benchmarks (section 4.1). Our minimal scheme (that should be applicable in non-visual domains) consists of Gaussian noise (with $\sigma = 0.1$) added to the pixel values. The standard scheme (indicated by 'TF' in Table 1) was used by Laine & Aila (2017) and adds translations in the interval $[-2, 2]$ and horizontal flips for the CIFAR-10 ↔ STL experiments. The affine scheme (indicated by 'TFA') adds random affine transformations defined by the matrix in (1), where $\mathcal{N}(0, 0.1)$ denotes a real value drawn from a normal distribution with mean 0 and standard deviation 0.1.

$$\begin{bmatrix} 1 + \mathcal{N}(0, 0.1) & \mathcal{N}(0, 0.1) \\ \mathcal{N}(0, 0.1) & 1 + \mathcal{N}(0, 0.1) \end{bmatrix} \tag{1}$$

The use of translations and horizontal flips has a significant impact in a number of our benchmarks. It is necessary in order to outpace prior art in the MNIST ↔ USPS and SVHN → MNIST benchmarks and improves performance in the CIFAR-10 ↔ STL benchmarks. The use of affine augmentation can improve performance in experiments involving digit and traffic sign recognition datasets, as seen in the 'MT+CT+TFA' row of Table 1. In contrast it can impair performance when used with photographic datasets, as seen in the the STL → CIFAR-10 experiment. It also impaired performance in the VisDA-17 experiment (section 4.2).

## 3.4 CLASS BALANCE LOSS

With the adaptations made so far the challenging MNIST → SVHN benchmark remains undefeated due to training instabilities. During training we noticed that the error rate on the SVHN test set decreases at first, then rises and reaches high values before training completes. We diagnosed the problem by recording the predictions for the SVHN target domain samples after each epoch. The rise in error rate correlated with the predictions evolving toward a condition in which most samples are predicted as belonging to the '1' class; the most populous class in the SVHN dataset. We hypothesize that the class imbalance in the SVHN dataset caused the unsupervised loss to reinforce the '1' class more often than the others, resulting in the network settling in a degenerate local minimum. Rather than distinguish between digit classes as intended it seperated MNIST from SVHN samples and assigned the latter to the '1' class.

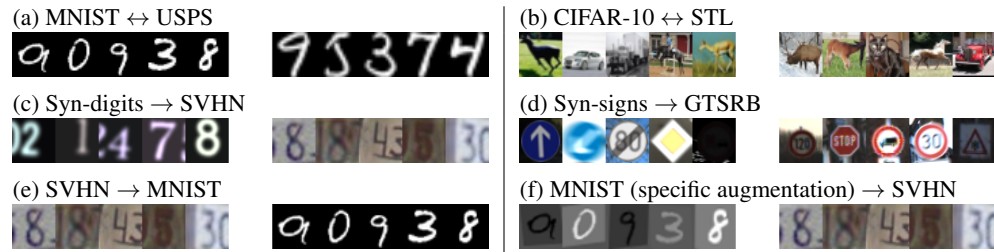

Figure 3: Small image domain adaptation example images

We addressed this problem by introducing a class balance loss term that penalises the network for making predictions that exhibit large class imbalance. For each target domain mini-batch we compute the mean of the predicted sample class probabilities over the sample dimension, resulting in the mini-batch mean per-class probability. The loss is computed as the binary cross entropy between the mean class probability vector and a uniform probability vector. We balance the strength of the class balance loss with that of the self-ensembling loss by multiplying the class balance loss by the average of the confidence threshold mask (e.g. if 75% of samples in a mini-batch pass the confidence threshold, then the class balance loss is multiplied by 0.75).[2]

We would like to note the similarity between our class balance loss and the entropy maximisation loss in the IMSAT clustering model of Hu et al. (2017); IMSAT employs entropy maximisation to encourage uniform cluster sizes and entropy minimisation to encourage unambiguous cluster assignments.

## 4 EXPERIMENTS

Our implementation was developed using PyTorch (Chintala et al.) and is publically available at http://github.com/Britefury/self-ensemble-visual-domain-adapt.

### 4.1 SMALL IMAGE DATASETS

Our results can be seen in Table 1. The 'train on source' and 'train on target' results report the target domain performance of supervised training on the source and target domains. They represent the expected baseline and best achievable result. The 'Specific aug.' experiments used data augmentation specific to the MNIST → SVHN adaptation path that is discussed further down.

The small datasets and data preparation procedures are described in Appendix A. Our training procedure is described in Appendix B and our network architectures are described in Appendix D. The same network architectures and augmentation parameters were used for domain adaptation experiments and the supervised baselines discussed above. It is worth noting that only the training sets of the small image datasets were used during training; the test sets used for reporting scores only.

**MNIST ↔ USPS** (see Figure 3a). MNIST and USPS are both greyscale hand-written digit datasets. In both adaptation directions our approach not only demonstrates a significant improvement over prior art but nearly achieves the performance of supervised learning using the target domain ground truths. The strong performance of the base mean teacher model can be attributed to the similarity of the datasets to one another. It is worth noting that data augmentation allows our 'train on source' baseline to outpace prior domain adaptation methods.

**CIFAR-10 ↔ STL** (see Figure 3b). CIFAR-10 and STL are both 10-class image datasets, although we removed one class from each (see Appendix A.2). We obtained strong performance in the STL → CIFAR-10 path, but only by using confidence thresholding. The CIFAR-10 → STL results are more interesting; the 'train on source' baseline performance outperforms that of a network trained on the STL target domain, most likely due to the small size of the STL training set. Our self-ensembling results outpace both the baseline performance and the 'theoretical maximum' of a network trained

---

[2]We expect that class balance loss is likely to adversely affect performance on target datasets with large class imbalance.

| | USPS – MNIST | MNIST – USPS | SVHN – MNIST | MNIST – SVHN | CIFAR – STL | STL – CIFAR | Syn Digits – SVHN | Syn Signs – GTSRB |
|---|---|---|---|---|---|---|---|---|
| **TRAIN ON SOURCE** | | | | | | | | |
| SupSrc[*] | 77.55 ±0.8 | 82.03 ±1.16 | 66.5 ±1.93 | 25.44 ±2.8 | 72.84 ±0.61 | 51.88 ±1.44 | 86.86 ±0.86 | 96.95 ±0.36 |
| SupSrc+TF | 77.53 ±4.63 | 95.39 ±0.93 | 68.65 ±1.5 | 24.86 ±3.29 | 75.2 ±0.28 | 59.06 ±1.02 | 87.45 ±0.65 | 97.3 ±0.16 |
| SupSrc+TFA | 91.97 ±2.15 | 96.25 ±0.54 | 71.73 ±5.73 | 28.69 ±1.59 | 75.18 ±0.76 | 59.38 ±0.58 | 87.16 ±0.85 | 98.02 ±0.20 |
| Specific aug.[b] | – | – | – | 61.99 ±3.9 | – | – | – | – |
| RevGrad[a] [1] | 74.01 | 91.11 | 73.91 | 35.67 | 66.12 | 56.91 | 91.09 | 88.65 |
| DCRN [2] | 73.67 | 91.8 | 81.97 | 40.05 | 66.37 | 58.65 | – | – |
| G2A [3] | 90.8 | 92.5 | 84.70 | 36.4 | – | – | – | – |
| ADDA [4] | 90.1 | 89.4 | 76.00 | – | – | – | – | – |
| ATT [5] | – | – | 86.20 | 52.8 | – | – | 93.1 | 96.2 |
| SBADA-GAN [6] | 97.60 | 95.04 | 76.14 | 61.08 | – | – | – | – |
| ADA [7] | – | – | 97.6 | – | – | – | 91.86 | 97.66 |
| **OUR RESULTS** | | | | | | | | |
| MT+TF | 98.07 ±2.82 | **98.26** ±0.11 | 99.18 ±0.12 | 13.96[c] ±4.41 | 80.08 ±0.25 | 18.3 ±9.03 | 15.94 ±0.0 | 98.63 ±0.09 |
| MT+CT[*] | 92.35 ±8.61 | 88.14 ±0.34 | 93.33 ±5.88 | 33.87[c] ±4.02 | 77.53 ±0.11 | 71.65 ±0.67 | 96.01 ±0.08 | 98.53 ±0.15 |
| MT+CT+TF | 97.28 ±2.74 | 98.13 ±0.17 | 98.64 ±0.42 | 34.15[c] ±3.56 | 79.73 ±0.45 | **74.24** ±0.46 | 96.51 ±0.08 | 98.66 ±0.12 |
| MT+CT+TFA | **99.54** ±0.04 | 98.23 ±0.13 | **99.26** ±0.05 | 37.49[c] ±2.44 | **80.09** ±0.31 | 69.86 ±1.97 | **97.11** ±0.04 | **99.37** ±0.09 |
| Specific aug.[b] | – | – | – | **97.0**[c] ±0.06 | – | – | – | – |
| **TRAIN ON TARGET** | | | | | | | | |
| SupTgt[*] | 99.53 ±0.02 | 97.29 ±0.2 | 99.59 ±0.08 | 95.7 ±0.13 | 67.75 ±2.23 | 88.86 ±0.38 | 95.62 ±0.2 | 98.49 ±0.32 |
| SupTgt+TF | 99.62 ±0.04 | 97.65 ±0.17 | 99.61 ±0.04 | 96.19 ±0.1 | 70.98 ±0.79 | 89.83 ±0.39 | 96.18 ±0.09 | 98.64 ±0.09 |
| SupTgt+TFA | 99.62 ±0.03 | 97.83 ±0.17 | 99.59 ±0.06 | 96.65 ±0.11 | 70.03 ±1.13 | 90.44 ±0.38 | 96.59 ±0.09 | 99.22 ±0.22 |
| Specific aug.[b] | – | – | – | 97.16 ±0.05 | – | – | – | – |

[1] Ganin & Lempitsky (2015), [2] Ghifary et al. (2016), [3] Sankaranarayanan et al. (2017), [4] Tzeng et al. (2017), [5] Saito et al. (2017a), [6] Russo et al. (2017), [7] Haeusser et al. (2017)
[a] RevGrad results were available in both Ganin & Lempitsky (2015) and Ghifary et al. (2016); we drew results from both papers to obtain results for all of the experiments shown.
[b] MNIST → SVHN specific intensity augmentation as described in Section 4.1.
[c] MNIST → SVHN experiments used class balance loss.

Table 1: Small image benchmark classification accuracy; each result is presented as *mean ± standard deviation*, computed from 5 independent runs. The abbreviations for components of our models are as follows: MT = mean teacher, CT = confidence thresholding, TF = translation and horizontal flip augmentation, TFA = translation, horizontal flip and affine augmentation, * indicates minimal augmentation.

on the target domain, lending further evidence to the view of Sajjadi et al. (2016) and Laine & Aila (2017) that self-ensembling acts as an effective regulariser.

**Syn-Digits → SVHN** (see Figure 3c). The Syn-Digits dataset is a synthetic dataset designed by Ganin & Lempitsky (2015) to be used as a source dataset in domain adaptation experiments with SVHN as the target dataset. Other approaches have achieved good scores on this benchmark, beating

the baseline by a significant margin. Our result improves on them, reducing the error rate from 6.9% to 2.9%; even slightly outpacing the 'train on target' 3.4% error rate achieved using supervised learning.

**Syn-Signs → GTSRB** (see Figure 3d). Syn-Signs is another synthetic dataset designed by Ganin & Lempitsky (2015) to target the 43-class GTSRB (German Traffic Signs Recognition Benchmark; Stallkamp et al. (2011)) dataset. Our approach halved the best error rate of competing approaches. Once again, our approaches slightly outpaces the 'train on target' supervised learning upper bound.

**SVHN → MNIST** (see Figure 3e). Google's SVHN (Street View House Numbers) is a colour digits dataset of house number plates. Our approach significantly outpaces other techniques and achieves an accuracy close to that of supervised learning.

**MNIST → SVHN** (see Figure 3f). This adaptation path is somewhat more challenging as MNIST digits are greyscale and uniform in terms of size, aspect ratio and intensity range, in contrast to the variably sized colour digits present in SVHN. As a consequence, adapting from MNIST to SVHN required additional work. Class balancing loss was necessary to ensure training stability and additional experiment specific data augmentation was required to achieve good accuracy. The use of translations and affine augmentation (see section 3.3) results in an accuracy score of 37%. Significant improvements resulted from additional augmentation in the form of random intensity flips (negative image), and random intensity scales and offsets drawn from the intervals $[0.25, 1.5]$ and $[-0.5, 0.5]$ respectively. These hyper-parameters were selected in order to augment MNIST samples to match the intensity variations present in SVHN, as illustrated in Figure 3f. With these additional modifications, we achieve a result that significantly outperforms prior art and nearly achieves the accuracy of a supervised classifier trained on the target dataset. We found that applying these additional augmentations to the source MNIST dataset only yielded good results; applying them to the target SVHN dataset as well yielded a small improvement but was not essential. It should also be noted that this augmentation scheme raises the performance of the 'train on source' baseline to just above that of much of the prior art.

## 4.2 VISDA-2017 VISUAL DOMAIN ADAPTATION CHALLENGE

The VisDA-2017 image classification challenge is a 12-class domain adaptation problem consisting of three datasets: a training set consisting of 3D renderings of sketchup models, and validation and test sets consisting of real images (see Figure 1) drawn from the COCO Lin et al. (2014) and YouTube BoundingBoxes Real et al. (2017) datasets respectively. The objective is to learn from labeled computer generated images and correctly predict the class of real images. Ground truth labels were made available for the training and validation sets only; test set scores were computed by a server operated by the competition organisers.

While the algorithm is that presented above, we base our network on the pretrained ResNet-152 (He et al. (2016)) network provided by PyTorch (Chintala et al.), rather than using a randomly initialised network as before. The final 1000-class classification layer is removed and replaced with two fully-connected layers; the first has 512 units with a ReLU non-linearity while the final layer has 12 units with a softmax non-linearity. Results from our original competition submissions and newer results using two data augmentation schemes are presented in Table 2. Our reduced augmentation scheme consists of random crops, random horizontal flips and random uniform scaling. It is very similar to scheme used for ImageNet image classification in He et al. (2016). Our competition configuration includes additional augmentation that was specifically designed for the VisDA dataset, although we subsequently found that it makes little difference. Our hyper-parameters and competition data augmentation scheme are described in Appendix C.1. It is worth noting that we applied test time augmentation (we averaged predictions form 16 differently augmented images) to achieve our competition results. We present resuts with and without test time augmentation in Table 2. Our VisDA competition test set score is also the result of ensembling the predictions of 5 different networks.

## 5 CONCLUSIONS

We have presented an effective domain adaptation algorithm that has achieved state of the art results in a number of benchmarks and has achieved accuracies that are almost on par with traditional supervised learning on digit recognition benchmarks targeting the MNIST and SVHN datasets. The

| VALIDATION PHASE | | TEST PHASE | |
|---|---|---|---|
| Team / model | Mean class acc. | Team / model | Mean class acc. |
| OTHER TEAMS | | | |
| bchidlovski [1] | **83.1** | NLE-DA [1] | 87.7 |
| BUPT_OVERFIT | 77.8 | BUPT_OVERFIT | 85.4 |
| Uni. Tokyo MIL [2] | 75.4 | Uni. Tokyo MIL [2] | 82.4 |
| OUR COMPETITION RESULTS | | | |
| ResNet-50 model | 82.8[a] | ResNet-152 model | **92.8**[ab] |
| OUR NEWER RESULTS (all using ResNet-152) | | | |
| Minimal aug.[*] | 74.2 ±0.86 | Minimal aug.[*] | 77.52 ±0.78 |
| Reduced aug. | 85.4 ±0.2 | Reduced aug. | 91.17 ±0.17 |
| + test time aug. | **86.6** ±0.18[a] | + test time aug. | 92.25 ±0.21[a] |
| Competition config. | 84.29 ±0.24 | Competition config. | 91.14 ±0.14 |
| + test time aug. | 85.52 ±0.29[a] | + test time aug. | 92.41 ±0.15[a] |

[1] Csurka et al. (2017), [2] Saito et al. (2017b)
[a] Used test-time augmentation; averaged predictions of 16 differently augmentations versions of each image
[b] Our competition submission ensembled predictions from 5 independently trained networks

Table 2: VisDA-17 performance, presented as *mean ± std-dev* of 5 independent runs. Full results are presented in Tables 4 and 5 in Appendix C.

resulting networks will exhibit strong performance on samples from both the source and target domains. Our approach is sufficiently flexible to be usable for a variety of network architectures, including those based on randomly initialised and pre-trained networks.

Miyato et al. (2017) stated that the self-ensembling methods presented by Laine & Aila (2017) – on which our algorithm is based – operate by label propagation. This view is supported by our results, in particular our MNIST → SVHN experiment. The latter requires additional intensity augmentation in order to sufficiently align the dataset distributions, after which good quality label predictions are propagated throughout the target dataset. In cases where data augmentation is insufficient to align the dataset distributions, a pre-trained network may be used to bridge the gap, as in our solution to the VisDA-17 challenge. This leads us to conclude that effective domain adaptation can be achieved by first aligning the distributions of the source and target datasets – the focus of much prior art in the field – and then refining their correspondance; a task to which self-ensembling is well suited.

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

## A  DATASETS AND DATA PREPARATION

### A.1  SMALL IMAGE DATASETS

The datasets used in this paper are described in Table 3.

|  | # train | # test | # classes | Target | Resolution | Channels |
|---|---|---|---|---|---|---|
| USPS[a] | 7,291 | 2,007 | 10 | Digits | $16 \times 16$ | Mono |
| MNIST | 60,000 | 10,000 | 10 | Digits | $28 \times 28$ | Mono |
| SVHN | 73,257 | 26,032 | 10 | Digits | $32 \times 32$ | RGB |
| CIFAR-10 | 50,000 | 10,000 | 10 | Object ID | $32 \times 32$ | RGB |
| STL[b] | 5,000 | 8,000 | 10 | Object ID | $96 \times 96$ | RGB |
| Syn-Digits[c] | 479,400 | 9,553 | 10 | Digits | $32 \times 32$ | RGB |
| Syn-Signs | 100,000 | – | 43 | Traffic signs | $40 \times 40$ | RGB |
| GTSRB | 32,209 | 12,630 | 43 | Traffic signs | *varies* | RGB |

[a] Available from `http://statweb.stanford.edu/~tibs/ElemStatLearn/datasets/zip.train.gz` and `http://statweb.stanford.edu/~tibs/ElemStatLearn/datasets/zip.test.gz`
[b] Available from `http://ai.stanford.edu/~acoates/stl10/`
[c] Available from Ganin's website at `http://yaroslav.ganin.net/`

Table 3: datasets

### A.2  DATA PREPARATION

Some of the experiments that involved datasets described in Table 3 required additional data preparation in order to match the resolution and format of the input samples and match the classification target. These additional steps will now be described.

**MNIST $\leftrightarrow$ USPS** The USPS images were up-scaled using bilinear interpolation from $16 \times 16$ to $28 \times 28$ resolution to match that of MNIST.

**CIFAR-10 $\leftrightarrow$ STL** CIFAR-10 and STL are both 10-class image datasets. The STL images were down-scaled to $32 \times 32$ resolution to match that of CIFAR-10. The 'frog' class in CIFAR-10 and the 'monkey' class in STL were removed as they have no equivalent in the other dataset, resulting in a 9-class problem with 10% less samples in each dataset.

**Syn-Signs $\rightarrow$ GTSRB** GTSRB is composed of images that vary in size and come with annotations that provide region of interest (bounding box around the sign) and ground truth classification. We extracted the region of interest from each image and scaled them to a resolution of $40 \times 40$ to match those of Syn-Signs.

**MNIST $\leftrightarrow$ SVHN** The MNIST images were padded to $32 \times 32$ resolution and converted to RGB by replicating the greyscale channel into the three RGB channels to match the format of SVHN.

## B  SMALL IMAGE EXPERIMENT TRAINING

### B.1  TRAINING PROCEDURE

Our networks were trained for 300 epochs. We used the Adam Kingma & Ba (2015) gradient descent algorithm with a learning rate of 0.001. We trained using mini-batches composed of 256 samples, except in the Syn-digits → SVHN and Syn-signs → GTSRB experiments where we used 128 in order to reduce memory usage. The self-ensembling loss was weighted by a factor of 3 and the class balancing loss was weighted by 0.005. Our teacher network weights $t_i$ were updated so as to be an exponential moving average of those of the student $s_i$ using the formula $t_i = \alpha t_{i-1} + (1-\alpha)s_i$, with a value of 0.99 for $\alpha$. A complete pass over the target dataset was considered to be one epoch in all experiments except the MNIST → USPS and CIFAR-10 → STL experiments due to the small size of the target datasets, in which case one epoch was considered to be a pass over the larger soure dataset.

We found that using the proportion of samples that passed the confidence threshold can be used to drive early stopping (Prechelt (1998)). The final score was the target test set performance at the epoch at which the highest confidence threshold pass rate was obtained.

## C  VISDA-17

### C.1  HYPER-PARAMETERS

Our training procedure was the same as that used in the small image experiments, except that we used $160 \times 160$ images, a batch size of 56 (reduced from 64 to fit within the memory of an nVidia 1080-Ti), a self-ensembling weight of 10 (instead of 3), a confidence threshold of 0.9 (instead of 0.968) and a class balancing weight of 0.01. We used the Adam Kingma & Ba (2015) gradient descent algorithm with a learning rate of $10^{-5}$ for the final two randomly initialized layers and $10^{-6}$ for the pre-trained layers. The first convolutional layer and the first group of convolutional layers (with 64 feature channels) of the pre-trained ResNet were left unmodified during training.

Reduced data augmentation:

- scale image so that its smallest dimension is 176 pixels, then randomly crop a $160 \times 160$ section from the scaled image
- *No* random affine transformations as they increase confusion between the car and truck classes in the validation set
- random uniform scaling in the range $[0.75, 1.333]$
- horizontal flipping

Competition data augmentation adds the following in addition to the above:

- random intensity/brightness scaling in the range $[0.75, 1.333]$
- random rotations, normally distributed with a standard deviation of $0.2\pi$
- random desaturation in which the colours in an image are randomly desaturated to greyscale by a factor between 0% and 100%
- rotations in colour space, around a randomly chosen axes with a standard deviation of $0.05\pi$
- random offset in colour space, after standardisation using parameters specified by PyTorch implementation of ResNet-152

## D  NETWORK ARCHITECTURES

Our network architectures are shown in Tables 6 - 8.

|  | Plane | Bicycle | Bus | Car | Horse | Knife | |
|---|---|---|---|---|---|---|---|
| COMPETITION RESULTS | | | | | | | |
| ResNet-50 | 96.3 | 87.9 | 84.7 | 55.7 | 95.9 | 95.2 | |
| NEWER RESULTS (ResNet-152) | | | | | | | |
| Minimal aug | 92.94 ±0.52 | 84.88 ±0.73 | 71.56 ±3.08 | 41.24 ±1.01 | 88.85 ±1.31 | 92.40 ±1.14 | |
| Reduced aug | 96.19 ±0.17 | 87.83 ±1.62 | 84.38 ±0.92 | 66.47 ±4.53 | 96.07 ±0.28 | 96.06 ±0.62 | |
| + test time aug | 97.13 ±0.18 | 89.28 ±1.45 | 84.93 ±1.09 | 67.67 ±4.66 | 96.54 ±0.36 | 97.48 ±0.43 | |
| Competition config. | 95.93 ±0.29 | 87.36 ±1.19 | 85.22 ±0.86 | 58.56 ±1.81 | 96.23 ±0.18 | 95.65 ±0.60 | |
| + test time aug | 96.89 ±0.32 | 89.06 ±1.24 | 85.51 ±0.83 | 59.73 ±1.96 | 96.59 ±0.13 | 97.55 ±0.48 | |

|  | M.cycle | Person | Plant | Sk.brd | Train | Truck | **Mean Class Acc.** |
|---|---|---|---|---|---|---|---|
| COMPETITION RESULTS | | | | | | | |
| ResNet-50 | 88.6 | 77.4 | 93.3 | 92.8 | 87.5 | 38.2 | 82.8 |
| NEWER RESULTS (ResNet-152) | | | | | | | |
| Minimal aug | 67.51 ±1.79 | 63.46 ±1.72 | 84.47 ±1.22 | 71.84 ±5.40 | 83.22 ±0.73 | 48.09 ±1.41 | 74.20 ±0.86 |
| Reduced aug | 90.49 ±0.27 | 81.45 ±0.90 | 95.27 ±0.36 | 91.48 ±0.76 | 87.54 ±1.16 | 51.60 ±2.35 | 85.40 ±0.20 |
| + test time aug | 90.99 ±0.37 | 83.33 ±0.91 | 96.12 ±0.32 | 94.69 ±0.71 | 88.53 ±1.20 | 52.54 ±2.82 | 86.60 ±0.24 |
| Competition config. | 90.60 ±1.08 | 80.03 ±1.23 | 94.79 ±0.35 | 90.77 ±0.65 | 88.42 ±0.87 | 47.90 ±2.16 | 84.29 ±0.24 |
| + test time aug | 91.00 ±1.17 | 81.59 ±1.20 | 95.58 ±0.38 | 94.29 ±0.63 | 89.28 ±0.85 | 49.21 ±2.26 | 85.52 ±0.29 |

Table 4: Full VisDA-17 validation set results

| | Plane | Bicycle | Bus | Car | Horse | Knife | |
|---|---|---|---|---|---|---|---|
| COMPETITION RESULTS (ensemble of 5 models) | | | | | | | |
| ResNet-152 | 96.9 | 92.4 | 92.0 | 97.2 | 95.2 | 98.8 | |
| NEWER RESULTS (ResNet-152) | | | | | | | |
| Minimal aug | 88.44 ±1.37 | 84.80 ±1.81 | 75.08 ±1.63 | 84.08 ±2.28 | 79.95 ±1.93 | 72.62 ±7.98 | |
| Reduced aug | 95.63 ±0.61 | 89.90 ±0.64 | 91.44 ±0.34 | 96.18 ±0.63 | 94.17 ±0.25 | 96.51 ±0.41 | |
| + test time aug | 96.72 ±0.59 | 91.67 ±0.73 | 92.21 ±0.45 | 96.41 ±0.65 | 94.72 ±0.21 | 98.03 ±0.40 | |
| Competition config. | 95.13 ±0.39 | 90.09 ±0.37 | 91.21 ±0.82 | 96.94 ±0.34 | 94.39 ±0.48 | 96.87 ±0.33 | |
| + test time aug | 96.48 ±0.31 | 91.96 ±0.38 | 91.92 ±0.65 | 97.22 ±0.36 | 95.12 ±0.52 | 98.44 ±0.13 | |

| | M.cycle | Person | Plant | Sk.brd | Train | Truck | **Mean Class Acc.** |
|---|---|---|---|---|---|---|---|
| COMPETITION RESULTS (ensemble of 5 models) | | | | | | | |
| ResNet-152 | 86.3 | 75.3 | 97.7 | 93.3 | 94.5 | 93.3 | 92.8 |
| NEWER RESULTS (ResNet-152) | | | | | | | |
| Minimal aug | 63.60 ±1.55 | 56.59 ±1.73 | 95.40 ±0.52 | 73.79 ±5.43 | 77.57 ±1.76 | 78.33 ±3.12 | 77.52 ±0.78 |
| Reduced aug | 85.02 ±0.83 | 71.31 ±0.97 | 97.35 ±0.49 | 91.11 ±1.05 | 92.42 ±0.46 | 93.03 ±0.36 | 91.17 ±0.17 |
| + test time aug | 85.40 ±1.08 | 73.19 ±0.86 | 97.84 ±0.45 | 93.53 ±0.71 | 93.31 ±0.35 | 93.91 ±0.39 | 92.25 ±0.21 |
| Competition config. | 85.12 ±1.30 | 70.78 ±1.53 | 97.22 ±0.19 | 90.39 ±0.64 | 93.18 ±0.49 | 92.38 ±0.52 | 91.14 ±0.14 |
| + test time aug | 85.75 ±1.20 | 74.06 ±1.69 | 97.77 ±0.16 | 92.91 ±0.45 | 94.21 ±0.52 | 93.09 ±0.44 | 92.41 ±0.15 |

Table 5: Full VisDA-17 test set results

| Description | Shape |
|---|---|
| $28 \times 28$ Mono image | $28 \times 28 \times 1$ |
| Conv $5 \times 5 \times 32$, batch norm | $24 \times 24 \times 32$ |
| Max-pool, 2x2 | $12 \times 12 \times 32$ |
| Conv $3 \times 3 \times 64$, batch norm | $10 \times 10 \times 64$ |
| Conv $3 \times 3 \times 64$, batch norm | $8 \times 8 \times 64$ |
| Max-pool, 2x2 | $4 \times 4 \times 64$ |
| Dropout, 50% | $4 \times 4 \times 64$ |
| Fully connected, 256 units | 256 |
| Fully connected, 10 units, softmax | 10 |

Table 6: MNIST $\leftrightarrow$ USPS architecture

| Description | Shape |
|---|---|
| $32 \times 32$ RGB image | $32 \times 32 \times 3$ |
| Conv $3 \times 3 \times 128$, pad 1, batch norm | $32 \times 32 \times 128$ |
| Conv $3 \times 3 \times 128$, pad 1, batch norm | $32 \times 32 \times 128$ |
| Conv $3 \times 3 \times 128$, pad 1, batch norm | $32 \times 32 \times 128$ |
| Max-pool, 2x2 | $16 \times 16 \times 128$ |
| Dropout, 50% | $16 \times 16 \times 128$ |
| Conv $3 \times 3 \times 256$, pad 1, batch norm | $16 \times 16 \times 256$ |
| Conv $3 \times 3 \times 256$, pad 1, batch norm | $16 \times 16 \times 256$ |
| Conv $3 \times 3 \times 256$, pad 1, batch norm | $16 \times 16 \times 256$ |
| Max-pool, 2x2 | $8 \times 8 \times 256$ |
| Dropout, 50% | $8 \times 8 \times 256$ |
| Conv $3 \times 3 \times 512$, pad 0, batch norm | $6 \times 6 \times 512$ |
| Conv $1 \times 1 \times 256$, batch norm | $6 \times 6 \times 256$ |
| Conv $1 \times 1 \times 128$, batch norm | $6 \times 6 \times 128$ |
| Global pooling layer | $1 \times 1 \times 128$ |
| Fully connected, 10 units, softmax | 10 |

Table 7: MNIST $\leftrightarrow$ SVHN, CIFAR-10 $\leftrightarrow$ STL and Syn-Digits $\rightarrow$ SVHN architecture

| Description | Shape |
|---|---|
| $40 \times 40$ RGB image | $40 \times 40 \times 3$ |
| Conv $3 \times 3 \times 96$, pad 1, batch norm | $40 \times 40 \times 96$ |
| Conv $3 \times 3 \times 96$, pad 1, batch norm | $40 \times 40 \times 96$ |
| Conv $3 \times 3 \times 96$, pad 1, batch norm | $40 \times 40 \times 96$ |
| Max-pool, 2x2 | $20 \times 20 \times 96$ |
| Dropout, 50% | $20 \times 20 \times 96$ |
| Conv $3 \times 3 \times 192$, pad 1, batch norm | $20 \times 20 \times 192$ |
| Conv $3 \times 3 \times 192$, pad 1, batch norm | $20 \times 20 \times 192$ |
| Conv $3 \times 3 \times 192$, pad 1, batch norm | $20 \times 20 \times 192$ |
| Max-pool, 2x2 | $10 \times 10 \times 192$ |
| Dropout, 50% | $10 \times 10 \times 192$ |
| Conv $3 \times 3 \times 384$, pad 1, batch norm | $10 \times 10 \times 384$ |
| Conv $3 \times 3 \times 384$, pad 1, batch norm | $10 \times 10 \times 384$ |
| Conv $3 \times 3 \times 384$, pad 1, batch norm | $10 \times 10 \times 384$ |
| Max-pool, 2x2 | $5 \times 5 \times 384$ |
| Dropout, 50% | $5 \times 5 \times 384$ |
| Global pooling layer | $1 \times 1 \times 384$ |
| Fully connected, 43 units, softmax | 43 |

Table 8: Syn-signs $\rightarrow$ GTSRB architecture

