# OpenReview forum: "Self-ensembling for visual domain adaptation"
_ICLR.cc/2018/Conference — Accept (Poster)_

### Official Review · AnonReviewer2 · 2017-11-27

**Rating:** 7
**Confidence:** 4

**Review:**

This paper presents a domain adaptation algorithm based on the self-ensembling method proposed by [Tarvainen & Valpola, 2017]. The main idea is to enforce the agreement between the predictions of the teacher and the student classifiers on the target domain samples while training the student to perform well on the source domain. The teacher network is simply an exponential moving average of different versions of the student network over time.

Pros:
+ The paper is well-written and easy to read
+ The proposed method is a natural extension of the mean teacher semi-supervised learning model by [Tarvainen & Valpola, 2017]
+ The model achieves state-of-the-art results on a range of visual domain adaptation benchmarks (including top performance in the VisDA17 challenge)

Cons:
- The model is tailored to the image domain as it makes heavy use of the data augmentation. That restricts its applicability quite significantly. I’m also very interested to know how the proposed method works when no augmentation is employed (for fair comparison with some of the entries in Table 1).
- I’m not particularly fond of the engineering tricks like confidence thresholding and the class balance loss. They seem to be essential for good performance and thus, in my opinion, reduce the value of the main idea.
- Related to the previous point, the final VisDA17 model seems to be engineered too heavily to work well on a particular dataset. I’m not sure if it provides many interesting insights for the scientific community at large.

In my opinion, it’s a borderline paper. While the best reported quantitative results are quite good, it seems that achieving those requires a significant engineering effort beyond just applying the self-ensembling idea.

Notes:
* The paper somewhat breaks the anonymity of the authors by mentioning the “winning entry in the VISDA-2017”. Maybe it’s not a big issue but in my opinion it’s better to remove references to the competition entry.
* Page 2, 2.1, line 2, typo: “stanrdard” -> “standard”

Post-rebuttal revision:
After reading the authors' response to my review, I decided to increase the score by 2 points. I appreciate the improvements that were made to the paper but still feel that this work a bit too engineering-heavy, and the title does not fully reflect what's going on in the full pipeline.

---

> ### Author Response · Authors · 2017-12-31
> **Thank you for your review and feedback**
>
> Thank you for your review
>
> * We agree that our work is tailored to the image domain. With a view to addressing your concerns, we have run further experiments to quantify the effects of each part of our approach - including data augmentation - for all of the small image benchmarks. We have therefore removed Table 2 as the information that it presented can be more compactly shown in Table 1, alongside everything else. We have added further discussion of the effect of our affine augmentation to section 3.3 and demonstrated its effect on both domain adaptation and plain supervised experiments. What we currently have is the same augmentation scheme used by Lain et al. and Tarvainen at al, which consists of translations (all) and horizontal flips (CIFAR/STL only). Experiments with minimal augmentation (gaussian noise added to the input only, therefore usable outside the image domain) are currently running; we will add them if the experiments complete on time.
>
> Furthermore, we found that our model performs slightly better on the MNIST <-> SVHN experiments when using RGB images rather than greyscale, so we have replaced our greyscale results with RGB ones. This represents a slightly bigger domain jump, so we hope that this increases your confidence in our work.
>
> * We see what you mean concerning engineering tricks. In defence of confidence thresholding, rather than being a new additional trick it replaces a time-dependent ramp-up curve used by Laine et al. in their work. We have made this a little more explicit in section 3.3. As for class balancing loss, it is similar in purpose and implementation to the entropy maximisation loss used in the IMSAT model of Hu et al. (an unsupervised clustering model that also uses data augmentation). We have mentioned this in section 3.4. We did not cite this paper in our original version as we were unaware of it at the time.
>
> * We have run further VisDA experiments. We found that pairing back our augmentation scheme improved performance on the validation set and made little different on the test set. Our original complex augmentation scheme was tested on a very small subset (1280 samples) of the training and validation sets during the development of our model. It turns out that these results did not generalise to the full set, so lesson learned (we were facing a tight competition deadline too). Our new reduced augmentation scheme consists of random crops, random horizontal flips and random uniform scaling, thus bringing it in line with augmentation schemes commonly used in ImageNet networks, such as He et al.'s ResNets. We have also performed 5 independent runs of each of our newer experiments and given a breakdown of the results.

---

> > ### Comment · AnonReviewer2 · 2018-01-14
> > **Comment**
> >
> > Thank you for you comments! Regarding class balancing loss, I'm wondering if it's safe to force the predictions on the target batch to be similar to the uniform distribution. As you mention in the paper, SVHN is a non-balanced dataset therefore a random batch won't really follow the uniform label distribution. I guess one has to be very careful with the scale of that term.

---

> > > ### Author Response · Authors · 2018-01-15
> > > **Good point**
> > >
> > > Thanks for pointing this out, as its' most likely correct; problems will likely arise in situations where there is severe class imbalance in the target dataset.
> > >
> > > If the editors permit it, we may need to add this caveat to our paper.

---

### Official Review · AnonReviewer1 · 2017-11-29
**The paper authors domain adaptation problems using techniques from semi-supervised learning and achieves impressive empirical results.**

**Rating:** 7
**Confidence:** 3

**Review:**

The paper addresses the problem of domain adaptation: Say you have a source dataset S of labeled examples and you have a target dataset T of unlabeled examples and you want to label examples from the target dataset.

The main idea in the paper is to train two parallel networks, a 'teacher network' and a 'student network', where the student network has a loss term that takes into account labeled examples and there is an additional loss term coming from the teacher network that compares the probabilities placed by the two networks on the outputs. This is motivated by a similar network introduced in the context of semi-supervised learning by Tarvainen and Valpola (2017). The parameters are then optimized by gradient descent where the weight of the loss-term associated with the unsupervised learning part follows a Gaussian curve (with time). No clear explanation is provided for why this may be a good thing to try. The authors also use other techniques like data augmentation to enhance their algorithms.

The experimental results in the paper are quite nice. They apply the methodology to various standard vision datasets with noticeable improvements/gains and in one case by including additional tricks manage to better than other methods for VISDA-2017 domain adaptation challenge. In the latter, the challenge is to use computer-generated labeled examples and use this information to label real photographic images. The present paper does substantially better than the competition for this challenge.

---

> ### Author Response · Authors · 2017-12-31
> **Thank you**
>
> Thank you for your review.
>
> We have clarified our discussion of the Gaussian curve based unsupervised loss scaling that was originally proposed by Laine et al. Beyond stating that the scaling function must ramp up slowly they don't discuss their choice of scaling function, so we present it as is. That said, we would propose replacing it with confidence thresholding, especially as it is more stable that Gaussian ramp-up in more challenging scenarios. We have explicitly clarified this in section 3.2.

---

### Official Review · AnonReviewer3 · 2017-11-29
**The method was not particularly novel but using "self-ensembling" seemed to win the VISDA 2017 domain adaptation competition.**

**Rating:** 7
**Confidence:** 5

**Review:**

The paper was very well-written, and mostly clear, making it easy to follow. The originality of the main method was not immediately apparent to me. However, the authors clearly outline the tricks they had to do to achieve good performance on multiple domain adaptation tasks: confidence thresholding, particular data augmentation, and a loss to deal with imbalanced target datasets, all of which seem like good tricks-of-the-trade for future work. The experimentation was extensive and convincing.

Pros:
* Winning entry to the VISDA 2017 visual domain adaptation challenge competition.
* Extensive experimentation on established toy datasets (USPS<>MNIST, SVHN<>MNIST, SVHN, GTSRB) and other more real-world datasets (including the VISDA one)

Cons:
* Literature review on domain adaptation was lacking. Recent CVPR papers on transforming samples from source to target should be referred to, one of them was by Shrivastava et al., Learning from Simulated and Unsupervised Images through Adversarial Training, and another by Bousmalis et al., Unsupervised Pixel-level Domain Adaptation with GANs. Also you might want to mention Domain Separation Networks which uses gradient reversal (Ganin et al.) and autoencoders (Ghifary et al.). There was no mention of MMD-based methods, on which there are a few papers. The authors might want to mention non-Deep Learning methods also, or that this review relates to neural networks,
* On p. 4 it wasn't clear to me how the semi-supervised tasks by Tarvainen and Laine were different to domain adaptation. Did you want to say that the data distributions are different? How does this make the task different. Having source and target come in different minibatches is purely an implementation decision.
* It was unclear to me what  footnote a. on p. 6 means. Why would you combine results from Ganin et al. and Ghifary et al. ?
* To preserve anonymity keep acknowledgements out of blind submissions. (although not a big deal with your acknowledgements)

---

> ### Author Response · Authors · 2017-12-31
> **Thank you for your review**
>
> Thank you for your review. We hope that our revision will address your concerns.
>
> * Thanks for pointing out the shortcomings of our literature review. We have stated that we are focusing on neural networks and we have cited the works that you mentioned. We have briefly mentioned MMD based approaches, although not in detail as we do not have an in-depth familiarity with the mathematics behind it. We have had to condense our literature review somewhat in order to not go too far over the page limit.
>
> * We have made the distinction between semi-supervised learning and domain adaptation more clear, as the distributions of the source and target datasets are indeed different. As for having separate source and target mini-batches, we have clarified how this fits in and was inspired by the work of Li et al. (2016). Time permitting, we may be able to run some experiments to quantify the effect this decision has and add the results to Table 1.
>
> * It seems that Ghifary et al. reimplemented Ganin's RevGrad approach. Neither paper had results for all the small image benchmarks that we discuss, so we took results from both papers to get a complete set. We have clarified the footnote.
>
> * We have suppressed the acknowledgements for now.

---

### Author Response · Authors · 2017-11-08
**Corrections to text in red**

There are three instances of text in red that indicate items that we would like to correct.

Firstly in the conclusions section on page 9 the word 'check' in red was a 'note to self' to verify the fact that our networks also exhibit strong performance on sample from the source domain. At submission time our experiment logs from our small image benchmarks backed this claim up. At the time we had not managed to verify this claim for the VisDA experiments, hence the 'note to self'. This has since been done and the claim holds. Given our approach to training (simultaneous supervised training on source domain and unsupervised training on target domain) we had a strong reason to believe this claim to be true at the time of submission.

In tables 1 and 2 on page 6 there are results in red, as they result from averaging less than the 5 independent runs as claimed in the table 1 caption. We have since run more experiments to get the full 5 results. The only substantial change is the 11.11 +/- 0 result for STL -> CIFAR in the 'Mean teacher' row which has now changed to 15.51 +/- 8.7. The rest are within a few tenths of a % of the results shown in the submitted version.

Furthermore, since submission we discovered a bug in our image augmentation code that affects the small colour image benchmarks (STL <-> CIFAR, Syn Digits -> SVHN and SynSigns -> GTSRB). Fixing the bug looks set to yield improved results (so far by looking at the results from the experiments that have completed). We would like to update tables 1 and 2 to reflect this.

---

### Decision · Program_Chairs · 2018-01-29
**ICLR 2018 Conference Acceptance Decision**

**Decision:**

Accept (Poster)

**Comment:**

An interesting application of self-ensembling/temporal ensembling for visual domain adaptation that achieves state of the art on the visual domain adaptation challenge. Reviewers noted that the approach is quite engineering-heavy, but I am not sure it's really much worse than making a pixel-to-pixel approach work well for domain adaptation.

I hope the authors follow through with their promise to add experiments to the final version (notably the minimal augmentation experiments to show just how much this domain adaptation technique is tailored towards imagenet-like things).

As it stands, this paper would be a good contribution to ICLR as it shows an efficient and interesting way to solve a particular visual domain adaptation problem.

---

> ### Author Response · Authors · 2018-02-04
> **Thank you**
>
> Thank you for accepting our paper! Would you like us to make the following changes to the paper?
>
> - The minimal augmentation results are in the paper for the small image experiments. We are running experiments using minimal augmentation on the VisDa dataset. Would you like us to insert these results into Table 2?
>
> - We could also add some (*) marks to table 1 to indicate minimal augmentation results more clearly if you like.
>
> - We also suggested to AnonReviewer2 that we could note a caveat concerning our class balancing loss.
>
> - We would also like to add web URLs for our source code.
>
> Thank you

---

> > ### Comment · Area_Chair · 2018-03-06
> > **proposed changes**
> >
> > Apologies for missing this message. Yes, the proposed changes sound good to me!